

# Identification of MOS9 as an interaction partner for chalcone synthase in the nucleus

Jonathan I. Watkinson[1], Peter A. Bowerman[1,2], Kevin C. Crosby[1,3], Sherry B. Hildreth[1,4], Richard F. Helm[4] and Brenda S.J. Winkel[1]

[1] Department of Biological Sciences, Virginia Polytechnic Institute and State University (Virginia Tech), Blacksburg, VA, USA
[2] BASF Plant Science LP, Research Triangle Park, NC, USA
[3] Department of Pharmacology, University of Colorado School of Medicine, Aurora, CO, USA
[4] Department of Biochemistry, Virginia Polytechnic Institute and State University (Virginia Tech), Blacksburg, VA, USA

## ABSTRACT

Plant flavonoid metabolism has served as a platform for understanding a range of fundamental biological phenomena, including providing some of the early insights into the subcellular organization of metabolism. Evidence assembled over the past three decades points to the organization of the component enzymes as a membrane-associated complex centered on the entry-point enzyme, chalcone synthase (CHS), with flux into branch pathways controlled by competitive protein interactions. Flavonoid enzymes have also been found in the nucleus in a variety of plant species, raising the possibility of alternative, or moonlighting functions for these proteins in this compartment. Here, we present evidence that CHS interacts with MOS9, a nuclear-localized protein that has been linked to epigenetic control of *R* genes that mediate effector-triggered immunity. Overexpression of *MOS9* results in a reduction of *CHS* transcript levels and a metabolite profile that substantially intersects with the effects of a null mutation in *CHS*. These results suggest that the MOS9–CHS interaction may point to a previously-unknown mechanism for controlling the expression of the highly dynamic flavonoid pathway.

## BACKGROUND

Flavonoids are well known as the distinctive red, blue, and purple pigments in flowers, fruits, and vegetables that have attracted substantial interest as beneficial phytochemicals and potential pharmaceuticals (*Mouradov & Spangenberg, 2014*; *Panche, Diwan & Chandra, 2016*; *Winkel-Shirley, 2001*). In plants, flavonoids play critical roles in reproduction and dispersal, protection from environmental stresses including UV radiation and cold, communication with other organisms, and modulation of auxin transport (*Gayomba, Watkins & Muday, 2017*; *Stafford, 1991*; *Winkel-Shirley, 2002*). Flavonoids have also been linked to plant defense, both as phytotoxins that are induced

Corresponding author
Brenda S.J. Winkel, winkel@vt.edu

in response to pathogen attack, and conversely, as metabolites that are specifically repressed as part of the plant immune response (*Serrano et al., 2012*).

Flavonoid biosynthesis has been extensively characterized at the biochemical, genetic, and molecular levels in a wide range of plant species and, while complex, is arguably among the best-understood systems of plant specialized metabolism. It has served as a highly-tractable system for the exploration of fundamental genetic, biological, and biochemical phenomena, including studies over the past 35 years on the organization of this pathway as an enzyme complex or "metabolon" (*Hrazdina & Jensen, 1992*; *Shih et al., 2008*; *Winkel-Shirley, 1999*; *Winkel, 2004*). The complex is characterized by relatively loose associations among its component parts, which may allow for rapid disassembly and reconstitution in response to cellular and environmental cues, with the distribution of flux further mediated by competition between key branchpoint enzymes for association with the entry-point enzyme, chalcone synthase (CHS), which appears to serve as a hub protein in this system (*Burbulis & Winkel-Shirley, 1999*; *Crosby et al., 2011*; *Owens et al., 2008a*, *2008b*). While there has long been evidence that this system is assembled at the cytoplasmic face of the endoplasmic reticulum, anchored by cytochrome P450 hydroxylases, there is growing evidence for the presence of flavonoid enzymes in the nucleus and perhaps other subcellular compartments, as well (reviewed in *Stafford, 1990*; *Winkel, 2006*, *in press*).

In the current study, we examined the possibility that CHS interacts with additional proteins that may direct the cellular localization, composition, and perhaps even activity of the flavonoid enzyme complex. Use of a yeast two-hybrid screen identified a novel interactor for CHS, a nuclear-localized protein identified by *Xia et al. (2013)* as MOS9, a regulator of *R* genes that are critical for plant defense. High resolution metabolite profiles of seedlings overexpressing this protein were consistent with a model in which interaction of MOS9 with CHS modifies flavonoid metabolism, either directly or indirectly. These findings suggest that the flavonoid enzyme complex may be part of a larger network of interacting proteins that connect the pathway's enzymes and endproducts to specific physiological functions, including plant defense, through previously-unknown mechanisms.

## METHODS

### Yeast two-hybrid assay

Chalcone synthase and chalcone isomerase (CHI) were previously cloned into pBI-880 as a fusion with the GAL4 DNA-binding domain (*Burbulis & Winkel-Shirley, 1999*). An Arabidopsis cDNA library in pBI-771 (*Pelaz et al., 2001*) was generously provided by William Crosby (University of Windsor, Canada). Screening was performed in Hf7c cells in the presence of five mM 3-amino-1,2,4-triazole (3-AT) as described in *Kohalmi et al. (1998)*. Two plates were screened, each containing ~$2 \times 10^7$ cells.

### Bioinformatics

Homologs of MOS9 and the protein encoded by At1g56420 were identified in other species using NCBI BLAST and Blink (*Sayers et al., 2012*). Protein sequences were aligned and a phylogenetic tree was constructed using the Clustal W algorithm in the MegAlign module of Lasergene (DNASTAR, Madison, WI, USA); bootstrapping was performed

using 1,000 trials. Protein structural comparisons were performed using 3D-Jury (http://bioinfo.pl/Meta/). Analysis and clustering of publicly-available microarray data was performed using the compendium-wide condition search tools in Genevestigator (*Zimmermann et al., 2004*). Secondary structures for MOS9 and the protein encoded by At1g56420 were predicted using the Protean module of Lasergene v. 8 (DNASTAR).

## Plant growth conditions

Seedlings for protein interaction, RNA expression, and localization analyses were grown on sterile medium in Nunc OmniTrays (Fisher, Hampton, NH, USA) containing 30 ml of 1× basal MS medium (Gibco, Gaithersburg, MD, USA) containing 30 mM (1.03%) sucrose and 0.7% Phytagel or Bacto agar (Sigma-Aldrich, St. Louis, MO, USA). Seeds were sterilized and distributed on the surface of the medium as described previously (*Kubasek et al., 1992*). Plates were wrapped in Nescofilm and incubated at four °C for 3 days before transferring to a growth chamber at 22 °C under 24 h light (~100 µE).

Seedlings for metabolite profiling were grown on sterile medium in 150 × 15 mm petri dishes containing 1× basal MS salts supplemented with 1% sucrose and 0.8% agar. Growth conditions were as described above except that plates were incubated for 2 days at four °C before transfer to the growth chamber. Whole seedlings were harvested at 5 days postgermination, frozen in liquid nitrogen, and stored at −80 °C prior to extraction.

## T-DNA insertion mutants

Seeds for the T-DNA insertional mutant, SAIL_622_D10, were obtained from the Arabidopsis Biological Resource Center (ABRC). Plants homozygous for the insertion were identified by polymerase chain reaction (PCR) analysis of genomic DNA extracted from flower buds using the method of *Edwards, Johnstone & Thompson (1991)* with the gene-specific primers to screen for intact genes and the SAIL LB1 and gene-specific primers to screen for T-DNA insertions (Table S1).

## Plasmid constructs

The *MOS9* coding region, minus the stop codon, was amplified from clone pENTR221-AT1G12530 (obtained from the ABRC, TAIR accession 1000491988) using the primers listed in Table S1. The amplified fragment was cloned into pENTR D-TOPO (Invitrogen, Carlsbad, CA, USA) according to the manufacturer's instructions to generate pENTR:MOS9. This construct was recombined with pET-DEST42 (Invitrogen, Carlsbad, CA, USA) using LR Clonase Plus Enzyme Mix (Invitrogen, Carlsbad, CA, USA) to give MOS9::V5::6×His for production of recombinant protein in *Escherichia coli* pENTR:MOS9 was also recombined into pEarleyGate 101 (*Earley et al., 2006*) using LR Clonase Plus to give *de35S::MOS9::YFP-HA* for stable transformation of Arabidopsis. The MOS9 construct for frequency domain fluorescence lifetime imaging microscopy-Förster resonance energy transfer (FLIM-FRET) experiments was generated by amplifying the *MOS9* coding region using the primers listed in Table S1, which was then inserted into the *Bam*HI site of syfp2-C1_pENTR (*Crosby et al., 2011*). The *MOS9-syfp2* sequences were then recombined into the plant expression vector,

p2BW7.0 (*Karimi, Inze & Depicker, 2002*), using LR Clonase II enzyme mix (Invitrogen, Carlsbad, CA, USA). The integrity of all constructs was confirmed by sequencing (performed at the Virginia Bioinformatics Institute Core Laboratory).

## Confirmation of protein interactions by SPR and FLIM-FRET

Recombinant proteins were produced in *E. coli* BL21 (DE3) pLysS cells (Invitrogen, Carlsbad, CA, USA) induced with one µM isopropyl thiogalactopyranoside for 4 h at room temperature. Proteins were extracted in 10 mM HEPES, pH7.5, 150 mM NaCl, 10% glycerol, 0.1% Tween20, and then purified by Fast Protein Liquid Chromatography (FPLC; ÄKTA, GE Healthcare, Chicago, IL, USA) over a one ml nickel fast flow column (Amersham, Louisville, CO, USA) using a linear gradient of 20–300 mM imidazole. Fractions containing the recombinant protein were combined and dialyzed against 10 mM HEPES, pH7.5, 150 mM NaCl to remove the imidazole. Immunoblot analysis of the resulting protein preparations showed that a horseradish peroxidase-conjugated rabbit anti-V5 antibody (Bethel Laboratories, Montgomery, TX, USA) reacted with a band of the predicted size.

Surface plasmon resonance refractometry (SPR) was performed on a SR7000 optical biosensor (Reichert Analytical Instruments, Depew, NY, USA) equipped with a syringe pump and autosampler. Planar polyethyleneglycol/carboxyl sensor chips (Reichert, Buffalo, NY, USA) were activated with 0.05M *N*-hydroxysuccinimide, 0.2M and 1-ethyl-3-(3-dimethylaminopropyl) carbodiimide injected at a flow rate of five µl/min for 10 min. Recombinant CHS (1.0 µM in 10 mM HEPES, pH7.5, 150 mM NaCl containing 20 mM sodium acetate, pH 5.0) was injected over the activated chip at five µl/min for 15 min. Unbound sites were deactivated with 1M ethanolamine, pH 8.0 for 15 min at five µl/min and unbound protein removed with a 10 min wash of 20 mM NaOH at 50 µl/min. MOS9 was diluted to 0.1–1µM concentration in running buffer (10 mM HEPES, pH7.5, 150 mM NaCl) with one mg/ml BSA as a blocking agent. Injections were passed over the chip at 50 µl/min for 8 min followed by 8 min dissociation in buffer without MOS9 and a 5 min wash in 20 mM NaOH to remove residual MOS9. All experiments were repeated three times with the order of injections randomized. Off rate ($K_{off}$) was calculated using Scrubber software (version 2.0, BioLogic Software, Campbell, Australia). The $k_{obs}$ was calculated using Origin software following a Langmuir binding plot. The association rate, $K_{on}$, was calculated from the slope of the line plotting $k_{obs}$ vs. concentration. The affinity ($K_D$) was calculated from $K_{off}/K_{on}$.

Frequency domain fluorescence lifetime imaging microscopy-Förster resonance energy transfer assays were performed as described in *Crosby et al. (2011)*. Mesophyll protoplasts generated from *tt4-11* (*Yoo, Cho & Sheen, 2007*) were transfected with the MOS9::scfp3a construct together with constructs for scfp3a, CHS::syfp2, or syfp2::CHS (*Crosby et al., 2011*). Between 11 and 25 protoplasts were analyzed for each donor- or donor/acceptor-expressing population. FRET efficiency was calculated using lifetime values derived from phase-shift, $\tau(\varphi)$, according to the formula: $E = 1-(\tau_{DA}/\tau_D)$ where $\tau_{DA}$ is the lifetime of the donor with the acceptor presence and $\tau_D$ is the lifetime of the donor without acceptor.

## Subcellular localization

Transformation with the *de35S::MOS9::YFP-HA* construct was performed using the floral dip method (*Bechtold, Ellis & Pelletier, 1993*). Transgenic seedlings were identified by screening for resistance to BASTA, followed by confirmation by PCR. T2 plants were used for localization analysis. For imaging of live seedlings, 5-day-old Arabidopsis were gently transferred to slides. Imaging was performed using a Zeiss LSM 510 Meta confocal laser scanning microscope (Carl-Zeiss, Oberkochen, Germany) using a Zeiss apochromat 40× water objective. Samples were excited with a 514 nm argon laser. Filter sets used were a HFT 458/514 dichroic beamsplitter and a BP 530–600 emission filter. Images were collected in eight-bit mode with four times line averaging. All images were collected using the same laser intensity, gain, offset, and filter settings. Stacks consisting of one µm intervals were taken to confirm the observed nuclear localization.

## Expression analysis and qRT-PCR

RNA was extracted from roots of approximately 20, 5-day-old Arabidopsis seedlings using an RNeasy RNA extraction kit (Qiagen, Hilden, Germany). Each set of 20 seedlings constituted one biological replicate. Biological replicates were grown on separate plates under the same conditions. cDNA was synthesized using 500 ng of total RNA from each sample as template with the Moloney Murine Leukemia Virus (MMLV) Reverse Transcriptase system according to the manufacturer's instruction (Epicentre, Charlotte, NC, USA).

Quantitative real-time polymerase chain reaction (qRT-PCR) analysis was performed using a 7,300 Real-Time PCR System (Applied Biosystems, Foster City, CA, USA) with SYBR Green PCR master mix (Applied Biosystems, Foster City, CA, USA). Primers were designed to assay the expression of *MOS9* and *CHS* as well as two control genes, encoding glyceraldehyde-3-phosphate dehydrogenase C-2 (*GAPDH*, At1g13440) and ubiquitin-conjugating enzyme 21 (*UBC*, At5G25760). Primers were designed to span introns where possible (*GAPDH* and *UBC*) to reduce the possibility of genomic contamination (Table S1). Primer efficiencies were determined by linear regression of Ct and log template concentration over a 5-log template concentration range. Efficiencies were calculated using the equation: $10^{(-1/\text{slope})} - 1$. Amplification of a single target was assessed by melting analysis and amplicon analysis via electrophoresis in 3% agarose gels. Three biological replicates were analyzed in triplicate using the Pfaffl Ct analysis method, normalizing to expression of *GAPDH* (*Pfaffl, 2001*). Given the possibility for variation of this control, a second control gene, *UBC*, was also assayed.

## Metabolite profiling

Whole seedlings were ground into a fine powder by agitation of two ml polypropylene tubes containing the frozen tissue and stainless steel beads (2.3 mm diameter; Small Parts, Logansport, IA, USA) in a Harbil 5G paint shaker (Fluid Management, Wheeling, IL, USA) for three 30 s cycles. The powder was extracted in 99% methanol (Spectrum Chemicals, New Brunswick, NJ, USA) with 1% acetic acid (Sigma-Aldrich, St. Louis, MO, USA) at a concentration of 100 mg tissue/ml, sonicated for 10 min, and then centrifuged to pellet the insoluble material. A second extraction was performed and the extracts were

combined. The resulting extract was dried under vacuum, reconstituted at a final concentration of 50 mg/ml in 75:25 water:acetonitrile with 0.1% formic acid, sonicated disregard for 10 min, and centrifuged to pellet the insoluble material.

Metabolite profiling was performed using a Waters Acquity I-class ultra performance liquid chromatography (UPLC) coupled with a Waters Synapt G2-S Q-TOF mass spectrometer (Waters Corp, Milford, MA, USA). Sample separation was achieved with a binary solvent system of water (Spectrum Chemicals, New Brunswick, NJ, USA) with 0.1% formic acid (Sigma-Aldrich, St. Louis, MO, USA) (A) and acetonitrile (Spectrum Chemicals, New Brunswick, NJ, USA) with 0.1% formic acid (B) at a flow rate of 0.2 ml/min using the following gradient conditions: hold at 5% B (0–1 min), linear gradient to 30% B (1–7 min), to 95% B (7–12 min), hold at 95% B (12–12.5 min), and then return to initial conditions and re-equilibrate (13–15 min). A two µl sample was injected onto an Acquity BEH C18 column (50 × 2.1 mm i.d., 1.7 µm; Waters Corp., Milford, MA, USA) maintained at 40 °C.

The column was eluted into the mass spectrometer fitted with an electrospray ionization (ESI) probe and analysis was performed in high resolution, negative mode with a scan time of 0.20 s and a mass scan range of 100–1,800 m/z. Source conditions were: capillary voltage 2.4, temperature 120 °C, cone voltage 30, source offset 80, desolvation temperature 350 °C, cone gas 50 l/h, desolvation gas 500 l/h, and nebulizer gas 6.0 bar. For accurate mass, leucine-enkephalin (Waters Corp., Milford, MA, USA) at a concentration of 200 ng/ml was continually infused through the reference sprayer at five µl/min.

The metabolomic data files were processed with MarkerLynx XS application manager software (version 4.1; Waters Corp., Milford, MA, USA) for unsupervised feature discovery, principal component analysis (PCA), and orthogonal partial least squared discriminant analysis (OPLS-DA) with processing parameters as detailed in Table S2. OPLS-DA was performed as a direct comparison of the metabolomic profiles of *de35S::MOS9::YFP* and Col-0 to identify feature differences highly correlated with the *de35S::MOS9::YFP* genotype. To determine the metabolomic relationship of *de35S::MOS9::YFP* with *tt4-11* and *mos9-2*, the fold change and *p*-value (calculated by Student's *t*-test) of the *de35S::MOS9::YFP* associated features were calculated for the *tt4-11* and *mos9-2* genotypes based on the MarkerLynx XS values.

Fragmentation analysis by MS/MS was performed to assign identities to the *de35S::MOS9::YFP*-associated features using the same chromatography and source conditions described above. Details related to fragmentation and identification of the metabolites are provided in Table S3. Identification of the flavonol conjugates was performed from MS$^E$ analysis, which allows for simultaneous detection of low and high collision energy data to provide untargeted fragmentation information. The flavonoid conjugates were identified based upon diagnostic fragment ions of the flavonols, kaempferol, quercetin, and isorhamnetin; the supporting data is provided in Table S4.

# RESULTS

## Identification of a novel interaction partner for CHS

Previous interaction assays suggest that CHS may serve as a hub for the assembly of a flavonoid enzyme complex (*Burbulis & Winkel-Shirley, 1999*; *Crosby et al., 2011*;
*Owens et al., 2008a*, *2008b*). To survey the Arabidopsis transcriptome for other potential interacting partners of CHS, a yeast two-hybrid screen was undertaken using a cDNA prey library generated from whole Arabidopsis plants at different stages of development (*Pelaz et al., 2001*). Screening was performed in Hf7c cells harboring a full-length CHS bait construct in the presence of five mM 3-AT. While a parallel experiment using a full-length CHI bait construct yielded no positives, the screen with CHS yielded approximately 30 colonies that grew to at least one mm in diameter after 6 days on selective medium. Of these, six exhibited robust growth after 10 days upon rescreening for histidine prototrophy in the presence of 3-AT. All six constructs tested negative for autoactivation of the His3 reporter gene in the absence of the CHS bait construct. β-galactosidase activity from the secondary reporter was not detected in any of the positive colonies; this was also the case in previous yeast two-hybrid analyses of flavonoid enzyme interactions, suggesting that these interactions are relatively weak (*Burbulis & Winkel-Shirley, 1999*).

One of the constructs contained sequences corresponding to the Arabidopsis gene, At1g12530, which encoded a small (193 amino acid, 21.8 kD) protein of as-yet unknown function. Although the coding region was out of frame with the Gal4 activation domain, translational frameshifting is known to occur at high frequency in yeast (*Albers et al., 2005*; *Fromont-Racine, Rain & Legrain, 1997*). The potential interaction of the two proteins was therefore examined further by SPR, using recombinant protein generated in *E. coli* from At1g12530 cDNA as the mobile analyte against immobilized recombinant CHS. Analysis of the binding isotherms showed a good fit between $k_{obs}$ vs. concentration (Fig. 1A) and gave a calculated $K_D$ of 210 nM.

To determine whether the interaction between CHS and the At1g12530 gene product could also be detected in vivo, the proteins were subjected to FLIM-FRET analysis (*Van Munster & Gadella, 2005*). The At1g12530 and CHS coding regions were fused to the optimized cyan and yellow fluorescent protein (YFP) variants, SCFP3A and SYFP2, respectively, and placed under control of the double enhanced CaMV 35S (de35S) promoter as described previously (*Crosby et al., 2011*). FLIM-FRET analysis in Arabidopsis mesophyll protoplasts showed no change in the fluorescence lifetime of the SCFP3A-At1g12530 protein when it was expressed in the presence of SYFP2 alone ($E\tau(\varphi)$ = 0.5%; Table 1). In contrast, a significant decrease in lifetime was observed when this protein was expressed together with CHS-SYFP2 ($E\tau(\varphi)$ = 6.5%; Table 1; Fig. 1B), similar to the efficiencies previously reported for interaction with two flavonoid enzymes, flavonol synthase 1 (FLS1), and dihydroflavonol 4-reductase (DFR) (*Crosby et al., 2011*). A substantially smaller effect on the lifetime of SCFP3A-At1g12530 was observed with SYFP2-CHS ($E\tau(\varphi)$ = 2.8%; Table 1) suggesting that fusion of the fluorescent protein to the *N*-terminus of CHS may interfere with the interaction or that dipole distance and/or that orientation for this pair of constructs is less favorable for FRET. Position-specific effects of the fusion partner were also observed in FLIM-FRET experiments for interactions between CHS, FLS1, and DFR (*Crosby et al., 2011*).

Taken together, these experiments provide strong evidence that CHS and the At1g12530 gene product interact in vivo. Remarkably, the product of this novel CHS interacting protein has been identified by *Xia et al. (2013)* as MOS9, a 193 amino acid

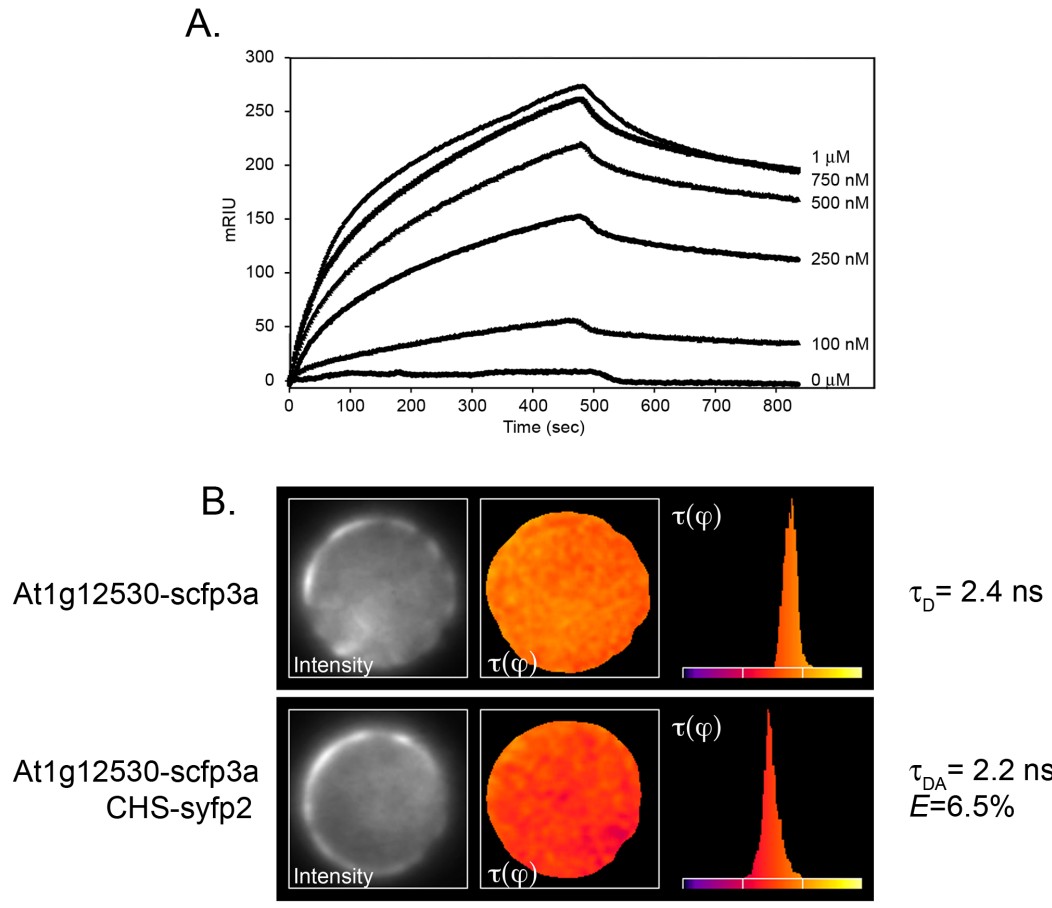

**Figure 1 Confirmation of the interaction of CHS with the At1g12530 protein.** (A) SPR analysis. Purified, recombinant CHS was immobilized on an SPR chip and purified, recombinant MOS9 passed over at the indicated concentrations. Each binding plot is shown as microresponse units (mRIU) over time and represents an average of three separate injections of MOS9. (B) FLIM-FRET analysis of At1g12530-SCFP3A and CHS-SYFP2 in Arabidopsis mesophyll protoplasts. *N*-terminal fusion constructs of either At1g12530 or CHS with cyan fluorescent protein and yellow fluorescent protein (respectively) were generated and used to transform living protoplasts. Fluorescence lifetime of At1g12530-SCFP3A alone was compared with its fluorescent lifetime in the presence of CHS-YFP2. Protoplasts are pseudo-colored to represent measured τ values. *E*-values for phase-shifts were calculated using *n* > 20 protoplasts (Table 1).               

(21.8 kD calculated) protein that associates with the Set1 class H3K4 methyl transferase, ATXR7, to activate transcription of *RPP4* and *SNC1* in response to pathogen effectors. Arabidopsis also contains a second related gene, At1g56420, which encodes a 183 amino acid (20.4 kD calculated) hypothetical protein with 32.3% amino acid identity to MOS9 (*Xia et al., 2013*), which we refer to as MOS9-L; it remains to be determined whether this protein has any of the same interaction partners. MOS9 and MOS9-L are both characterized by a high degree of predicted alpha helical structure, including amphipathic alpha helices (Fig. S1A). Like flavonoid enzymes, these proteins appear to be unique to the green lineage, with orthologs in other plant species and smaller versions (108–123 amino acids in length) encoded by the *Physcomitrella* (28.6–61.1% identity) and *Chlamydomonas* (15.6–32.5% identity) genomes (Fig. S1B).

**Table 1 FLIM-FRET parameters for analysis of the CHS-MOS9 interaction in Arabidopsis protoplasts[a].**

| Donor | Acceptor | $n$ | $\tau(\varphi)$ ns ± S.D | $\tau(m)$ ns | $E\ \tau(\varphi)$ |
|---|---|---|---|---|---|
| MOS9-scfp3a | – | 24 | 2.36 ± 0.07 | 3.27 ± 0.05 | – |
| MOS9-scfp3a | CHS-syfp2 | 25 | 2.21 ± 0.08 | 3.24 ± 0.08 | 6.5% |
| MOS9-scfp3a | syfp2-CHS | 21 | 2.29 ± 0.13 | 3.37 ± 0.07 | 2.8% |
| MOS9-scfp3a | – | 12 | 2.40 ± 0.05 | 3.26 ± 0.03 | – |
| MOS9-scfp3a | syfp2 | 11 | 2.41 ± 0.03 | 3.17 ± 0.05 | 0.5% |

**Note:**
[a] Results above and below the solid line are from independent experiments.

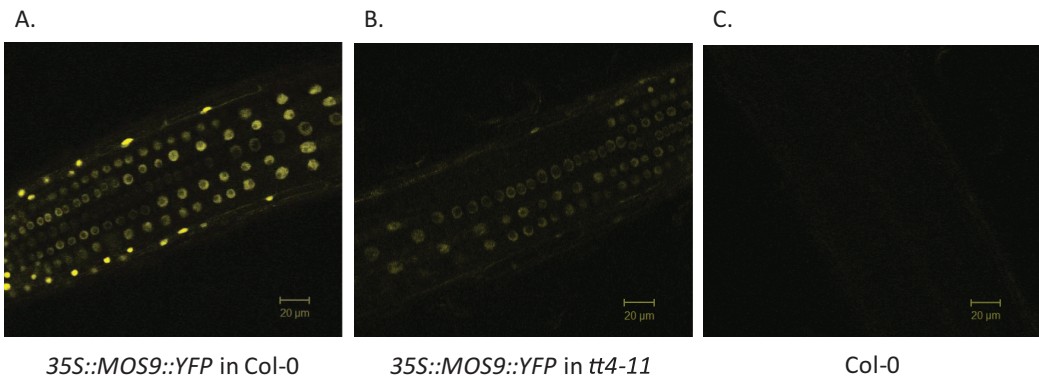

A.      B.     C.

*35S::MOS9::YFP* in Col-0     *35S::MOS9::YFP* in *tt4-11*     Col-0

**Figure 2 Localization of MOS9 in wild-type and CHS-deficient roots.** MOS9-YFP fusion proteins were expressed from the de35S promoter in stably-transformed wild-type Col-0 (A) and *tt4-11* plants (B). The root elongation zones of 5-day-old transgenic and Col-0 control (C) seedlings were examined by confocal laser scanning microscopy. All three images were taken under identical conditions and represent individual optical slices from stacks taken in the *z*-direction.

## Colocalization and coexpression of MOS9 and CHS

Chalcone synthase has been localized to both the cytoplasm, where it is to at least some extent associated with the endoplasmic reticulum, and to the nucleus (*Hrazdina & Wagner, 1985*; *Saslowsky, Warek & Winkel, 2005*; *Saslowsky & Winkel-Shirley, 2001*). *Xia et al. (2013)* showed that MOS9 is located predominantly in the nucleus of root and leaf pavement cells based on analysis of green fluorescent protein (GFP) fusions driven by the *MOS9* promoter in transgenic plants and cell fractionation experiments. We observed a similar distribution in Arabidopsis Col-0 roots for a MOS9-YFP fusion protein expressed under control of the de35S promoter (Figs. 2A and 2C). Predominantly nuclear localization was also the case in the absence of CHS, as evidenced from crossing one of these lines with the CHS null, *tt4-11,* and selecting for yellow-seeded plants expressing YFP (*Bowerman et al., 2012*) (Figs. 2B and 2C). The fluorescence of MOS9-YFP was considerably lower in the *tt4-11* background, even though the absence of flavonoids in these tissues should, if anything, provide for an enhanced fluorescence signal from YFP (*Mercuri et al., 2001*; *Robic, Lacorte & Miranda, 2009*). The finding that *MOS9* transcript

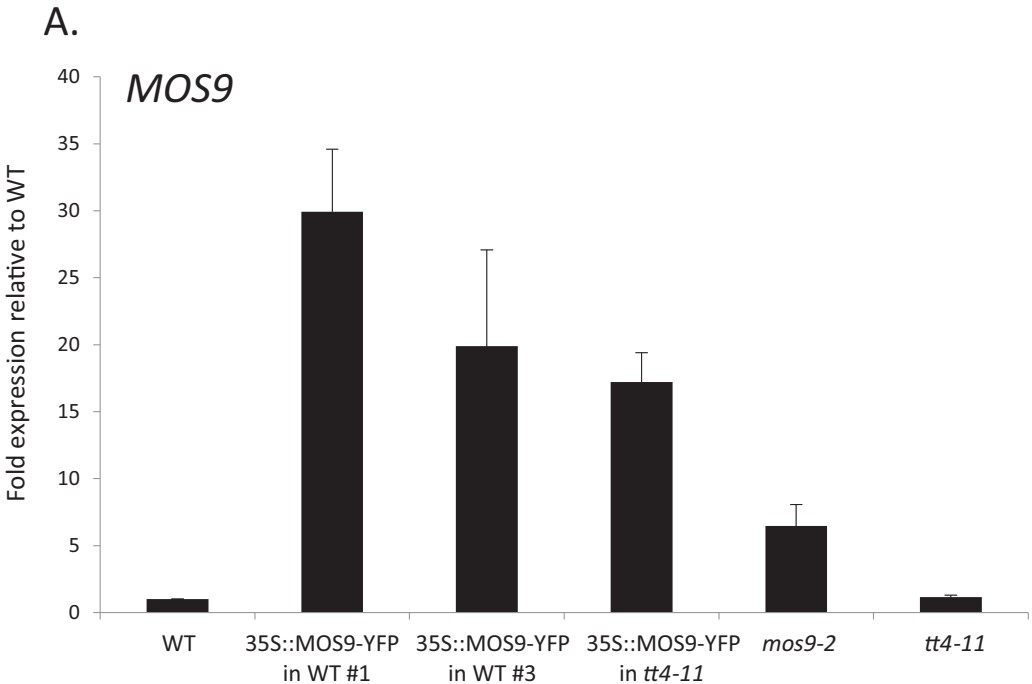

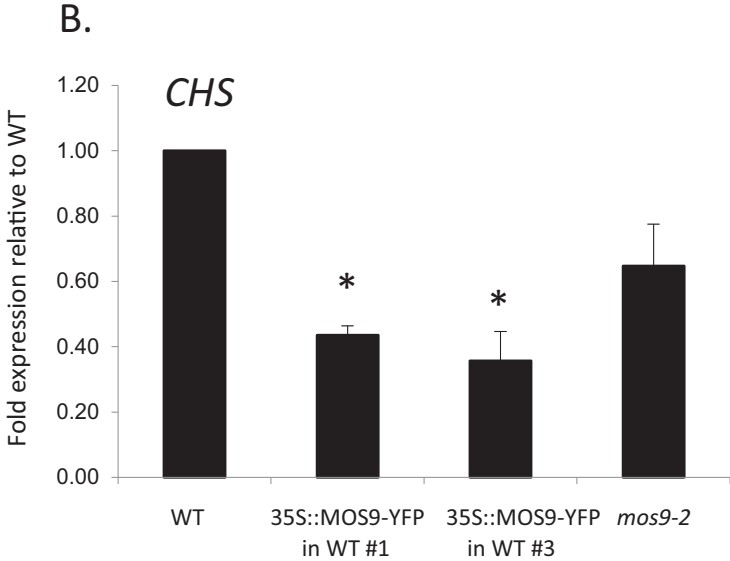

**Figure 3 Analysis of *MOS9* (A) and *CHS* (B) expression in roots of 5-day-old *MOS9* overexpression lines.** Transcript levels were analyzed by real-time PCR and normalized to the endogenous control gene, *GAPDH*. Error bars represent the standard error of the mean. Asterisks indicate a *p*-value < 0.02 from a two-tailed paired student's *t*-test for expression of *CHS* relative to wild type. WT = wild type (Col-0).

levels were not statistically different in this line (Fig. 3A) suggests that the stability of the MOS9 protein could be enhanced upon interaction with CHS.

The expression of *MOS9* was compared to that of *CHS* across different tissues and environmental conditions using Genevestigator (*Zimmermann et al., 2004*). This analysis indicates the both genes are expressed in almost all tissues, albeit with

somewhat distinct patterns of RNA accumulation (Fig. S2A). Both exhibit particularly high levels of expression in a variety of root and seed cell types. There are few tissues where only one of these genes is expressed, notably in the hypocotyl, where *MOS9* transcripts accumulate but *CHS* does not appear to be expressed. The two genes do exhibit markedly different responses to biotic and abiotic stress (Figs. S2B, S2C and S2D), with *MOS9* being relatively unresponsive across a range of perturbations, whereas *CHS* transcript levels change dramatically in plants exposed to diverse nutritional, environmental, and biotic stresses, as has also been well documented in the literature (reviewed in *Mouradov & Spangenberg, 2014*; *Winkel-Shirley, 2002*). This includes significantly ($p < 0.001$) reduced *CHS* transcript levels in response to several bacterial pathogens; *MOS9* displays a modest ($-1.73$-fold; $p < 0.05$) response only in leaf discs treated with the bacterial flagellin peptide, flg22, and gibberellic acid (where *CHS* is $-1.91$-fold, but with $p = 0.423$). Thus, while *CHS* and *MOS9* have the potential to act in concert in many, if not all, cell types, only *CHS* exhibits a strong stress response, including during pathogenesis.

## Genetic analysis

Although there are no publicly-available mutant lines with disruptions of the *MOS9* coding region, SAIL_622_D10 contains a T-DNA insertion located 165 bp upstream of the predicted start codon for this gene. Surprisingly, qRT-PCR analysis of expression in seedling roots showed that *MOS9* transcript levels, rather than being diminished, were approximately eightfold higher in this line than in wild-type when normalized to the endogenous control gene, *GAPDH* ($p < 0.02$) (Fig. 3A). Normalization to a second endogenous control gene, *UBC*, yielded similar results (Fig. S3A). The insertion thus likely disrupts a negative regulatory element in the *MOS9* promoter. Expression of the next closest annotated gene, At1g12540, which is divergently transcribed from a predicted start site located two kb upstream, was not affected by the insertion (Fig. S3B). The insertion line was therefore designated *mos9-2*. qRT-PCR also showed that *MOS9-YFP* transcript levels in the transgenic overexpression lines were variable, but consistently 10- to 30-fold above native *MOS9* in both wild-type and *tt4-11* backgrounds (Fig. 3A; Fig. S3A).

## Effects of *MOS9* on metabolism

Metabolite profiling was used to determine whether overexpression of *MOS9* affected plant metabolism, including the accumulation of flavonoids for which CHS activity is required. Profiles were obtained from 5-day-old *mos9-2, de35S::MOS9::YFP-1, de35S::MOS9::YFP-3, tt4-11,* and wild-type Col-0 seedlings by UPLC-ESI-MS. PCA analysis of the 502 metabolomic features showed clear separation of the genotypes (Fig. 4A), except for near superimposition of the two genotypically-similar *de35S::MOS9:: YFP* lines. Counter to expectations, the profile for *mos9-2* seedlings was not intermediate between wild-type Col-0 and the *de35S::MOS9::YFP* lines. This suggests that the modest increase in *MOS9* expression in *mos9-2* is not sufficient to mimic the effects on metabolism observed with overexpression in *de35S::MOS9::YFP*.

A.

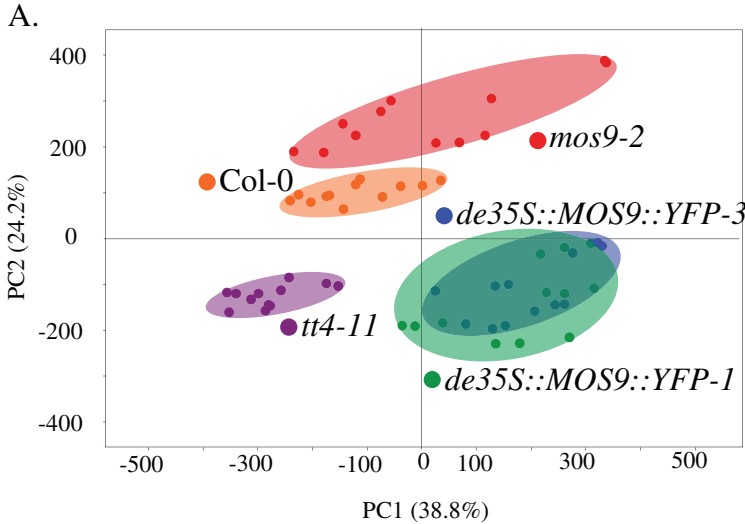

B.

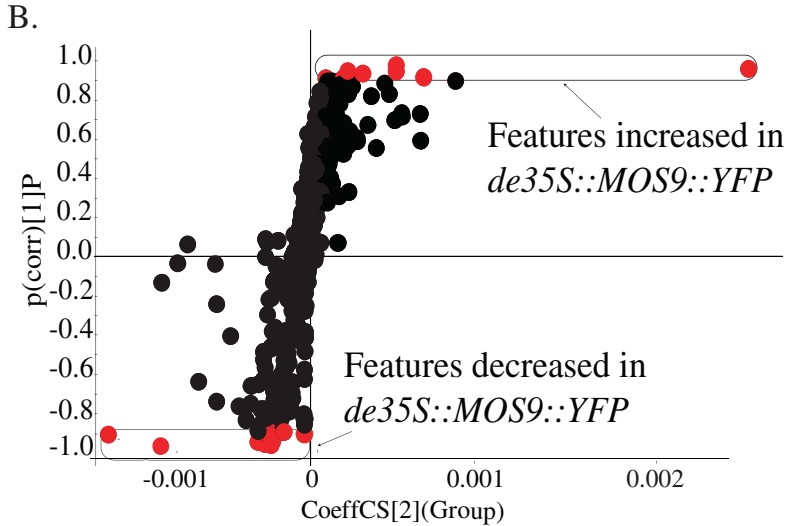

**Figure 4 Metabolite profiling of *de35S::MOS9::YFP, mos9-2, tt4-11,* and Col seedlings.** (A) Principal component analysis (PCA) of UPLC-MS profiles for four biological replicates of each genetic background. (B) *S*-plot distribution of features between *de35S::MOS9::YFP* and Col-0 seedlings. Features shown in red were found to be associated with the *MOS9* ectopic expression and were selected for further analysis.

A direct comparison of the profiles of *de35S::MOS9::YFP* and wild-type Col-0 is illustrated in the feature distribution plot (Fig. 4B). Of the 502 features generated by UPLC-MS analysis, 26 were tightly correlated with the *de35S::MOS9::YFP* genotype, with eight at higher levels and 18 at decreased levels relative to Col-0. After removal of redundant features derived from adducts, the 26 features were associated with 18 distinct metabolites, 11 of which could be assigned identities and were shown to fall into three distinct classes (Table 2; Table S3).

The largest group of metabolites correlated with overexpression of *MOS9* consisted of compounds derived from, or containing moieties derived from, phenylalanine. The most substantial changes, from 4.1- to 12.1-fold, were observed for the flavonol,

**Table 2 Metabolites with altered levels in *de35S::MOS9::YFP*, *mos9-2*, and *tt4-11* seedlings relative to wild-type Col-0.**

| Putative identity[a] | Retention time | Observed mass (adduct) | Ratio of change (*p*-value)[b,c] | | |
|---|---|---|---|---|---|
| | | | *de35S:: MOS9::YFP* | *mos9-2* | *tt4-11* |
| I. Phenylalanine derivatives | | | | | |
| Sinapoyl-glutamate | 4.52 | 352.1027 ([M-H]⁻) | **1.8 (<0.01)** | 1.1 (0.43) | **2.1 (<0.01)** |
| *N,N*-Bis(sinapoyl)-spermidine | 5.62 | 556.2656 ([M-H]⁻) | **2.5 (<0.01)** | 1.1 (0.53) | 1.1 (0.33) |
| 1,2-Di-*O*-sinapoyl-*β*-glucose | 6.40 | 591.1711 ([M-H]⁻) | **1.5 (<0.01)** | 1.1 (0.02) | 1.2 (0.29) |
| 4-Benzoyloxybutyl-glucosinolate | 5.77 | 494.0788 ([M-H]⁻) | **2.8 (<0.01)** | **1.5 (<0.01)** | **−1.4 (<0.01)** |
| Quercetin-Rhamnose | | | | | |
| Monomer | 5.92 | 447.0922 ([M-H]⁻) | **−4.1 (<0.01)** | −1.2 (0.03) | **n.d.** |
| Biflavonoid | 5.23 | 446.0843 ([M-2H]²⁻) | **−12.1 (<0.01)** | **−1.5 (<0.01)** | **n.d.** |
| Triflavonoid | 3.77 | 669.1270 ([M-2H]²⁻) | **−5.0 (<0.01)** | **−1.8 (<0.01)** | **n.d.** |
| Syringaresinol glucoside | 6.00 | 579.2068 ([M-H]⁻) | **−3.1 (<0.01)** | **1.5 (<0.01)** | **−1.5 (<0.01)** |
| II. Jasmonic acid precursors/Oxylipins | | | | | |
| Arabidopside A (isomer a) | 11.36 | 819.4523 ([M-H + HCO₂H]⁻) | **−2.9 (<0.01)** | 1.2 (0.35) | −1.1 (0.25) |
| Arabidopside A (isomer b) | 11.70 | 819.4523 ([M-H + HCO₂H]⁻) | **−2.9 (<0.01)** | 1.3 (0.09) | −1.1 (0.32) |
| Arabidopside C | 10.74 | 981.5046 ([M-H + HCO₂H]⁻) | **−5.0 (<0.01)** | 1.3 (0.33) | **−1.4 (<0.01)** |
| III. As-yet unidentified | | | | | |
| UK Glycerolipid 1 | 9.91 | 855.4731 ([M-H + HCO₂H]⁻) | **−3.1 (<0.01)** | 1.4 (0.07) | −1.0 (0.85) |
| UK Glycerolipid 2 | 10.16 | 855.4741 ([M-H + HCO₂H]⁻) | **−3.2 (<0.01)** | **1.4 (<0.01)** | −1.1 (0.66) |
| UK Glycerolipid 3 | 10.22 | 957.4696 ([M-H + HCO₂H]⁻) | **−2.6 (<0.01)** | 1.2 (0.22) | −1.2 (0.12) |
| UK Glycerolipid 4 | 10.38 | 767.3841 ([M-H + HCO₂H]⁻) | **−3.2 (<0.01)** | 1.0 (0.72) | −1.2 (0.10) |
| UK sulfur-containing | 4.35 | 307.0847 ([M-H]⁻) | **−6.4 (<0.01)** | **1.9 (<0.01)** | **−1.7 (<0.01)** |
| UK6 | 5.61 | 595.2021 ([M-H]⁻) | **−5.1 (<0.01)** | **1.7 (<0.01)** | **−1.4 (<0.01)** |
| UK7 | 10.41 | 783.3788 ([M-H + HCO₂H]⁻) | **−2.5 (<0.01)** | 1.1 (0.89) | **−1.3 (0.01)** |

**Notes:**
[a] UK, unknown; additional information on metabolite identification is presented in Table S1.
[b] Statistically-significant values are highlighted in bold type.
[c] n.d., not detected.

quercetin rhamnoside (4.1-fold lower), and its bi- (12.1-fold lower), and triflavonoid (5.0-fold higher) forms (*David et al., 2014*; *Routaboul et al., 2006*). The levels of these flavonoids were all slightly elevated in *mos9-2* and were entirely absent, as expected, in the CHS null mutant *tt4-11* (*Bowerman et al., 2012*). In addition to the *MOS9* correlated features identified in the feature distribution plot, a comparison of the flavonoids present in the metabolite profiles found small but significant changes in the levels of other flavonol glycosides in the three *MOS9* overexpression lines (Table S4). These profiles of altered and overall reduced flavonoid accumulation are consistent with the finding that *MOS9* overexpression reduces CHS transcript levels, as well as the possibility that interaction of MOS9 diverts CHS from its catalytic function. Additional statistically-significant changes (*p* < 0.01) within the group of phenylalanine-derived compounds included increased (1.5- to 2.5-fold) accumulation of three sinapoyl-containing compounds in the *MOS9* overexpression lines, one of which was also elevated in *tt4-11*. Two additional phenylpropanoid-containing compounds that are associated with defense functions exhibited statistically-significant changes in the overexpression lines,

with levels of 4-benzyloxybutyl glucosinolate elevated 2.8-fold, while syringaresinol glucoside was reduced 3.1-fold; these changes were somewhat smaller than many of the others and for that reason perhaps less consistent in *mos9-2* and *tt4-11*. Taken together, these observations provide further evidence that *MOS9* overexpression alters flux across phenylpropanoid metabolism, a hallmark of disruption of its branch pathways, even if the outcomes are not always easily predictable (*Anderson et al., 2015*; *Ring et al., 2013*).

A second group of metabolites associated with overexpression of *MOS9* consisted of arabidopsides, which are jasmonic acid precursors in the oxylipin family that have been associated with defense (*Griffiths, 2015*). Three of these compounds were present at approximately 3- to 5-fold lower levels in the *de35S::MOS9::YFP* lines than in Col-0 (Table 2). The third class was composed of compounds that could not be definitively identified, but included four glycerolipids; these metabolites were present 2.5- to 6.4-fold lower levels than in wild type plants. All of the compounds were also present at slightly reduced levels in *tt4-11,* although the changes were smaller and not all were statistically significant. Less clear were the changes in *mos9-2,* which were overall higher relative to Col-0, but statistically significant in only three cases. Overall, only the compounds with the largest changes in abundance in the *de35S::MOS9::YFP* lines were also appreciably altered in *mos9-2*. As above, the inconsistencies between these lines likely reflect the comparatively small change in the level of mRNA expression in the *mos9-2* mutant line.

### Effects of *MOS9* overexpression on CHS transcript levels

To determine whether *CHS* expression was affected by overexpression of *MOS9, CHS* transcript levels were assayed in roots of Col-0 wild type, *mos9-2*, and the two independent *de35S::MOS9::YFP* lines in the Col-0 background. qRT-PCR analysis indicated that over-expression of *MOS9* in seedling roots reduced the expression of *CHS* in a dose-dependent manner in seedling roots. *CHS* transcript levels were significantly lower in two independent *de35S::MOS9::YFP* lines, with an average expression of approximately 40% of wild-type levels ($p < 0.02$), and appeared to be somewhat reduced in the overexpression mutant, *mos9-2*, although with $p = 0.109$ (Fig. 3B; Fig. S3A). These results suggest that MOS9 influences *CHS* gene expression at the level of transcription and/or mRNA stability, leading to a decrease in CHS enzymatic activity, and thereby modulating the metabolite profile. The converse was not true, however, with *MOS9* transcripts at similar levels in the absence or presence of *CHS* (Fig. 3A; Fig. S3A).

### DISCUSSION

This study endeavored to identify new interaction partners for the key flavonoid enzyme, CHS, with the aim of adding to the catalog of components of an enzyme complex localized at the cytoplasmic face of the ER and perhaps also in the nucleus (*Winkel, 2006*). The interaction of Arabidopsis CHS with the At1g12530 gene product was initially deduced from a yeast two-hybrid screen and subsequently supported by SPR analysis of recombinant proteins produced in *E. coli*, indicating that the association does not rely on plant-specific post-translational modifications. FLIM-FRET analysis of coexpressed

fusion proteins further showed that the interaction can occur in the near-native environment of Arabidopsis mesophyll protoplasts. Overexpression of At1g12530 resulted in changes in the seedling metabolome that overlapped substantially with those observed in the *CHS* null mutant, *tt4-11*, consistent with a reduction in *CHS* transcripts in the overexpression lines. These findings, together with evidence for its subcellular localization, suggest that this newly-identified partner influences flavonoid metabolism and perhaps CHS gene expression through interaction with CHS protein in the nucleus.

Remarkably, AT1g12530 has been independently described as one of 15 *modifier of snc1* (*mos*) genes identified in a screen for suppressors of a constitutively-active allele of the NB-LRR-encoding resistance (R) gene, *snc1* (*suppressor of npr1-1 constitutive*) (*Monaghan et al., 2010*). R genes function to recognize pathogen effectors and activate effector-triggered immunity (ETI), a secondary defense mechanism that typically results in induction of the hypersensitive response (*Jones, Vance & Dangl, 2016*). The products of these 15 genes participate in a wide range of fundamental cellular processes, from nucleo-cytoplasmic trafficking to RNA processing and protein modification, all of which are also key components of the defense signaling network. *Xia et al. (2013)* subsequently showed that MOS9 is essential for expression of both *SNC1* and the adjacent NB-LRR R gene, *RPP4*. The action of MOS9 appears to involve association with ATXR7, a Set 1 class H3K4 histone methyltransferase that may also have a role in flowering time determination (*Yun et al., 2011*). MOS9 and ATXR7 were initially copurified via immunoprecipitation, but unlike the situation with MOS9 and CHS, additional interaction assays were unsuccessful. This led the authors to propose that MOS9 and ATXR7 may not interact directly, but rather constitute components of a larger complex that includes additional as-yet-to-be identified proteins that together mediate the expression of key defense signaling genes.

The new evidence that MOS9 interacts with CHS and that overexpression is associated with changes in *CHS* mRNA levels and the profile of phenolics in roots suggests that MOS9 may also play a role in controlling plant metabolism. One scenario is that MOS9 helps modulate *CHS* gene expression in response to cellular CHS protein levels as part of a negative feedback loop. The association of MOS9 with components of the R gene expression network suggests that it could also serve to coordinate multiple cellular processes, including during plant defense. Suppression of flavonoid gene expression and flavonoid levels is well known to accompany ETI, ostensibly allowing plants to redirect carbon flow into defense-related phenylpropanoid compounds such as scopoletin and lignin (*Schenke, Bottcher & Scheel, 2011*). It may also promote expression of defense-related genes such as late-response *PR-1* and *PR-2*, which appear to be repressed when flavonoids are present (*Serrano et al., 2012*). This suppression may involve histone remodeling of several flavonoid structural and regulatory genes, including CHS (*Schenke, Cai & Scheel, 2014*; *Velanis, Herzyk & Jenkins, 2016*; *Zhou et al., 2017*). One possibility is that the MOS9–CHS interaction either directly or indirectly impacts epigenetic modifications associated with these genes, analogous to its proposed role in control of R gene expression.

It should be noted that the metabolite profiles of MOS9 overexpressing and CHS null plants support the involvement of both proteins in additional, nonoverlapping, functions. Many of the metabolites that distinguish the *MOS9* overexpressing lines from wild type were also altered in *tt4-11*; overall, where there were substantial changes in the *35Sde::MOS9::YFP* lines, these were also observed in *mos9-2* and *tt4-11* (Table 2). This includes a number of phenylpropanoid and flavonoid compounds, consistent with CHS and MOS9 acting in, or on, the same metabolic pathways. However, the opposite effects on 4-benzoyloxybutyl glucosinolate levels suggest that MOS9 may also influences processes distinct from those stemming from interaction with CHS. The apparent constitutive expression of *MOS9* further suggests that, as with its involvement in control of *RPP4* and *SNC1* gene expression, there are surely other factors involved.

## CONCLUSIONS

Evidence for the interaction of the central flavonoid enzyme, CHS, with the nuclear-localized protein, MOS9, may have uncovered a novel mechanism for controlling flavonoid metabolism and linking this highly dynamic plant pathway with specific physiological functions, including the defense response. Identifying additional components of the MOS9 interaction network and elucidating the functions of the associated protein complexes will be key to shedding further light on this intriguing system.

## ACKNOWLEDGEMENTS

The authors are indebted to William L. Crosby of the University of Windsor for the gift of the yeast two-hybrid library, to Anna Pietraszewska-Bogiel and Dorus Gadella of the University of Amsterdam for providing the expertise and facilities for the FLIM-FRET experiments, and to Kristi DeCourcy at Virginia Tech for technical support with confocal microscopy.

### Funding

This work was supported by grants from the NSF Molecular Biochemistry (MCB-0445878), IGERT (DGE-0523658), and Arabidopsis 2010 (IOS-0820674) programs. The funders had no role in study design, data collection and analysis, decision to publish, or preparation of the manuscript.

### Grant Disclosures

The following grant information was disclosed by the authors:
NSF Molecular Biochemistry (MCB-0445878), IGERT (DGE-0523658), and Arabidopsis 2010 (IOS-0820674) programs.

### Competing Interests

Peter A. Bowerman is an employee of BASF Plant Science, LP. The authors declare that they have no competing interests.

## Author Contributions

- Jonathan I. Watkinson conceived and designed the experiments, performed the experiments, analyzed the data, contributed reagents/materials/analysis tools, prepared figures and/or tables, authored or reviewed drafts of the paper.
- Peter A. Bowerman conceived and designed the experiments, performed the experiments, analyzed the data, contributed reagents/materials/analysis tools, prepared figures and/or tables, authored or reviewed drafts of the paper.
- Kevin C. Crosby conceived and designed the experiments, performed the experiments, analyzed the data, contributed reagents/materials/analysis tools, prepared figures and/or tables, authored or reviewed drafts of the paper.
- Sherry B. Hildreth conceived and designed the experiments, performed the experiments, analyzed the data, contributed reagents/materials/analysis tools, prepared figures and/or tables, authored or reviewed drafts of the paper, approved the final draft.
- Richard F. Helm conceived and designed the experiments, analyzed the data, contributed reagents/materials/analysis tools, prepared figures and/or tables, authored or reviewed drafts of the paper, approved the final draft, provided financial support through grants and institutional funds.
- Brenda S.J. Winkel conceived and designed the experiments, analyzed the data, contributed reagents/materials/analysis tools, prepared figures and/or tables, authored or reviewed drafts of the paper, approved the final draft, provided financial support through grants.

## Data Availability

   The raw data are provided in Table S5.

## Supplemental Information

Supplemental information for this article can be found online at http://dx.doi.org/10.7717/peerj.5598#supplemental-information.

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
