# Peer review of "Identification of MOS9 as an interaction partner for chalcone synthase in the nucleus"

_PeerJ, doi:10.7717/peerj.5598_

## Round 0.1 · original submission · Major Revisions

The manuscript was well written yet the reviews were mixed, potentially requiring some additional inspection to help validate the conclusion. What appears to be a straightforward interaction may not be upon close inspection. It appears that many assumptions without validation are apparent and should be resolved in some fashion. The reviewers suggest approaches toward elucidating and confirming an interaction. Overall the reviewers comments are positive with critical reservations which should be addressed in a revision. There are several points suggested by the reviewers familiar with this area of work which should be addressed in a future revision. Thank you for your contribution and I look forward to seeing the updated manuscript.

Reviewer 1 ·

Basic reporting

There is no direct evidence that the interaction between MOS9 and CHS is related to the ETI response. The manuscript need to contain results relevant to the ETI response. Ideally, OX-MOS9 transgenic lines with tt4-11 background are inoculated with an incompatible pathogen and their disease symptoms are evaluated to test whether the interaction between MOS9 and CHI influences the ETI response.

Experimental design

Lines 292-294.
The comparison of the YFP fluorescence between MOS9 in WT and CHS-deficient roots needs a standard to evaluate difference in the fluorescence between them. For example, overexpression of another fluorescent protein such as CFP and RFP can be used as the standard. This is because differences in fluorescence between transgenic plant lines expressing the same fluorescent protein driven by the same promoter are commonly observed even when images are taken under identical conditions.

Validity of the findings

It is difficult to see that de35S::MOS9::YFP lines share substantial changes of the metabolites including phenylpropanoid and flavonoid compounds with mos9-2 and tt4-11 from Table 2 (lines 437-442). This is because de35S::MOS9::YFP lines and mos-9 have different signs (plus or minus) of ratio of change for syringaresinol hexoside, monomer and biflavonoid of quercetin rhamnoside, UNK glycerolipids in Table 2.

Additional comments

Coexpression of MOS9 and CHS was observed in the root tissues. If the hypothesis that the interaction is involved in the ETI response is correct, the interaction does not influence ETI response in the leaf, does not it? More explanations are needed.

Reviewer 2 ·

Basic reporting

Watkinson et al described the identification of a novel interaction partner of CHS, the nuclear localized protein MOS9 that potentially involved in regulation of effector-triggered immunity. The authors further observed that upregulation of MOS9 gene expression suppressed CHS gene expression and affect the accumulation of a set of phenylalanine-derived metabolites. Overall the manuscript is well written and nicely organized.

Experimental design

This is an explorative work, in which the authors intended to identify additional proteins interacting with CHS via yeast two hybrid screening, followed with SPR and FLIM-FRET analyses to validate the recognized interactions. Such experiments were designed properly and sound. The methods are clearly described.

Validity of the findings

While the discovery and confirmation of CHS-MOS interaction is convincing, the biological relevance of such protein-protein interaction seems less solidly elucidated. The authors observed the effects of overexpression of MOS9 on CHS transcript and the synthesis of a set of metabolites. But there is no any directly data support the observed protein-protein interaction of MOS9 and CHS has any link with transcriptional suppression of CHS. With such concern, I feel it is not so valid to conclude that “these findings …point to a role for this newly identified partner in suppressing CHS gene expression through interaction with CHS protein in the nucleus (Discussion Lines 405-406)”

Also the authors deduced that “the interaction of MOS9 diverts CHS from its cytosolic catalytic function”. To make this notion confirmatory, probably it needs more experimental data to support, e.g. the data on whether CHS’s subcellular localization could be quantitatively altered under co-expression of MOS9.

Regarding FLIM-FRET in mesophyll protoplast, it is not clear whether the FRET efficiency was calculated from the fluorescence within entire protoplast or in nucleus. It would be better that the authors could present the related fluorescent imaging data showing the co-localization of CHS and MOS9 (theoretically in nucleus).

Additional comments

Overall the manuscript describes a nice discovery, which could help to further understand the CHS- mediated protein complex and its potential interplay with other biological process

Reviewer 3 ·

Basic reporting

I believe that the style of this paper fulfills the Basic reporting criteria of the Journal.

Experimental design

Overall, the experimental designs described in this paper appear to fulfill the criteria of the Journal.

Validity of the findings

This paper reports on the finding concerning the interaction of CHS with a nuclear-localized protein (MOS9) that is linked to epigenetic control of genes mediating ETI. Although CHS has been known to be localized not only in cytoplasm but also in nucleus, physiological significance of its nuclear localization has been elusive. Therefore, the findings reported in this paper may provide useful information for insights into the possible role(s) of nuclear-localized CHS.

1. Line 352: .... overall reduced flavonol accumulation ...:
Table 2 shows ratios of changes without contents. Because triflavonoid showed 5.0-fold increase in de35S::MOS9::YFP, it cannot be concluded, without showing the contents, that total flavonols were reduced in de35S::MOS9::YFP, even though other flavonol species showed diminished changes. Please clarify.

2. Discussion section
The interaction of CHS and MOS9 has been unambiguously identified, but unfortunately the CHS transcription and metabolomics results obtained were only poorly discussed in terms of the observed CHS/MOS9 interaction. The reduced CHS transcript levels and the reduced flavonol levels observed upon MOS overexpression might be explained without considering the interaction of MOS9 with CHS but physiological significance of the interaction could be discussed.

2-1. The results reported show that CHS transcript levels were reduced upon the overexression of MOS9. This reviewer wonders whether this observation could be discussed in conjunction with the formation of the CHS/MOS9 complex. MOS9 appears to be constitutively expressed in plant (as mentioned on line 444) and was relatively unresponsive across a range of perturbations (line 302). Then what is the role of the CHS/MOS9 complex formation? Would it be possible to discuss such a scenario that excess CHS molecules in cytoplasm get into the nucleus and bind to the constitutive MOS9 to produce a small amount of the CHS/MOS9 complex, which in turn reduces the CHS transcript levels in a negative feedback manner?

2-2. If total flavonol glycosides indeed diminish upon the overexression of MOS9, this reviewer wonders whether it would be possible to elaborate/discuss this observation in terms of the formation of the CHS/MOS9 complex. For example, overproduction of MOS9 protein might result in the removal of CHS protein from cytoplasm (or ER-localized flavonoid metabolon) by incorporating it into nucleus.

2-3. Small amounts of flavonoids could be produced via reactions catalyzed by nuclear-localized flavonoid enzymes (including CHS) in the nucleus and exert some regulatory role(s) during gene expression events. Would it be possible that MOS9 plays a role to inhibit nuclear production of flavonoids by its binding to CHS? In this case, it would be interesting to examine whether CHS activity is inhibited by MOS9 binding. Do authors have any in vitro data concerning this?

Additional comments

Minor comment:

Table 2.
It would be better to express fold changes of reduction, for example, as 0.244 and 0.083 rather than –4.1 and –12.1, respectively. Please consider.

---

## Round 0.2 · accepted · Accept

Thank you for taking the time to address the comments made by the reviewers. I am satisfied that each was dealt with on the best possible terms. The manuscript was well written and includes a good presentation of the methods included. The manuscript provides a strong case for MOS9 and perhaps other possible interaction with CHS. The methods used and approach for determining interactions with targeted expression may be useful to model other such interactions. Th manuscript is ready to be moved forward and is acceptable for publication. Thank you for your contribution.

At line 77: you may want to add an affiliation in parenthesis after “William Crosby”, and another small item to address may be at line 83: “Blast and Blink” might be modified to “BLAST and BLink” to reflect consistency in the letter-case. There was a note that the BLink service was replaced, so perhaps a citation may be warranted here.

#